# ALBAR: Adversarial Learning approach to mitigate Biases in Action Recognition

**Joseph Fioresi, Ishan Rajendrakumar Dave, Mubarak Shah**
Center for Research in Computer Vision, University of Central Florida, Orlando, USA
`{joseph.fioresi, ishanrajendrakumar.dave}@ucf.edu, shah@crcv.ucf.edu`
`Project Page: https://joefioresi718.github.io/ALBAR_webpage/`

## Abstract

Bias in machine learning models can lead to unfair decision making, and while it has been well-studied in the image and text domains, it remains underexplored in action recognition. Action recognition models often suffer from background bias (i.e., inferring actions based on background cues) and foreground bias (i.e., relying on subject appearance), which can be detrimental to real-life applications such as autonomous vehicles or assisted living monitoring. While prior approaches have mainly focused on mitigating background bias using specialized augmentations, we thoroughly study both foreground and background bias. We propose ALBAR, a novel adversarial training method that mitigates foreground and background biases without requiring specialized knowledge of the bias attributes. Our framework applies an adversarial cross-entropy loss to the sampled static clip (where all the frames are the same) and aims to make its class probabilities uniform using a proposed *entropy maximization* loss. Additionally, we introduce a *gradient penalty* loss for regularization against the debiasing process. We evaluate our method on established background and foreground bias protocols, setting a new state-of-the-art and strongly improving combined debiasing performance by over **12%** absolute on HMDB51. Furthermore, we identify an issue of background leakage in the existing UCF101 protocol for bias evaluation which provides a shortcut to predict actions and does not provide an accurate measure of the debiasing capability of a model. We address this issue by proposing more fine-grained segmentation boundaries for the actor, where our method also outperforms existing approaches.

## 1 Introduction

In a wide range of computer vision tasks, models often rely on unintended and seemingly irrelevant patterns in the data, known as spurious correlations, as shortcuts to make predictions or decisions Geirhos et al. (2018; 2020). These correlations do not represent the true underlying relationship between the input features and the target output. As a result, models that exploit these spurious correlations may achieve high performance on the training and in-distribution test data, but fail to generalize well to real-world scenarios. A notable example of this is seen in video action recognition, where a model will choose an action label by only considering the background instead of the subjects motion Ding et al. (2022a); Zou et al. (2023). For example, if a subject is performing the action "Throwing Frisbee" while standing on a soccer field, a model will likely predict "Playing Soccer" instead. Here, the model is not using the motion information, instead using spatial information to make a biased decision. However, even if this background bias is mitigated, there may still be biases related to the video foreground Li et al. (2023). In our example, even if the subject is "Throwing Frisbee" indoors, but wearing a soccer jersey, a model may still erroneously predict "Playing Soccer". While the sources of foreground biases may include relatively harmless sources like a held object, they may also manifest in more harmful sources, such as a person's physical appearance attributes like skin color, facial hair etc. Zhao et al. (2017); Stock & Cisse (2017); Buolamwini & Gebru (2018); Wilson et al. (2019); Prabhu & Birhane (2020); Tong & Kagal (2020); Steed & Caliskan (2021); Gustafson et al. (2023). Such appearance-based decisions are highly undesirable in real-life applications of action recognition in security cameras, elderly monitor systems, or autonomous cars

where an unbiased decision is crucial. Despite extensive studies on background biases in action recognition, the area lacks comprehensive research on biases related to the foreground.

Adversarial learning has emerged as a promising method for debiasing neural network representations Beutel et al. (2017); Elazar & Goldberg (2018); Zhang et al. (2018); Wang et al. (2019). These techniques often require supplementary information, such as scene or object labels, or involve training a separate critic model to predict the biased attribute. For example, scene/object classifiers Duan et al. (2022) have been employed for mitigating scene bias in action recognition. The effectiveness of such methods is contingent upon the accuracy and reliability of the critic model, as they rely on its negative gradients for optimization. Since annotating video datasets with detailed attributes for biases requires enormous annotation efforts, it is not scalable. Li *et al.* Li et al. (2023) recently propose a method to mitigate bias in action recognition using augmentations based on MixUp Zhang et al. (2017), where they first detect action-salient frames from videos and add such frames to other videos. Their idea is to reduce static bias by encouraging the model not to make decisions based on the mixed frames. Although this method works well in reducing the background bias, it does not significantly reduce the foreground bias and relies on an off-the-shelf salient frame detector model.

To address these challenges, we propose a novel adversarial learning technique that eliminates the need for attribute labels or pretrained attribute classifiers and provides an end-to-end training framework. Like Bahng et al. (2020); Bao et al. (2021), we hypothesize that the bias issues in action recognition stem from an over-reliance on spatial information by the classifier. Prior works attempt to solve this by contrasting 3-dimensional (spatio-temporal) and 2-dimensional (spatial) representations using separate encoders. Instead, we break away from this formulation and design an adversarial framework based on a single 3D encoder model. Specifically, we first sample any frame from within a video clip and repeat it to obtain a static clip. To mitigate the static bias, we introduce an *adversarial loss* through negative gradients to penalize the model from making action-class predictions based on static cues and propose *entropy maximization* loss to make its class predictions uniform across all classes. We also introduce a *gradient penalty* objective to regularize the debiasing process. We term our method ALBAR (meaning in Arabic: "Guard of All", Adversarial Learning approach to mitigate Biases in Action Recognition), for which a schematic diagram is shown in Fig. 1.

We show state-of-the-art performance on a comprehensive video action recognition bias evaluation protocol Li et al. (2023) based on popular benchmarks such as Kinetics400 Carreira & Zisserman (2017), UCF101 Soomro et al. (2012), and HMDB51 Kuehne et al. (2011), namely SCUBA (static cues in the background) and SCUFO (static cues in the foreground). SCUBA evaluates background bias by replacing the background of action clips, and SCUFO evaluates foreground bias by stacking a single frame from SCUBA to create clips with no motion. We also found a shortcut in the prior UCF101 bias protocol Li et al. (2023), which used bounding boxes to separate the foreground from the background, allowing background information surrounding the bounded subject to leak into the protocol. We propose a fix to this version of the evaluation protocol that appropriately separates the foreground and background via segmentation masks.

The key contributions of this work can be outlined as follows:

- We propose a novel adversarial learning-based method to mitigate biases in action recognition, which provides simplified end-to-end training and does not require any labels/classifiers for bias-related attributes.
- Our adversarial learning framework consists of a negative-gradient-based loss paired with an entropy-maximization loss and a gradient norm penalty, which are combined to strongly discourage the model from making predictions based on static cues.
- Our method achieves strong state-of-the-art performance on established SCUBA/SCUFO background/foreground debiasing benchmarks: notably a strong $\approx$**12%** absolute increase in overall accuracy on the HMDB51 protocol.
- Having identified shortcuts in the prior UCF101 bias protocols due to background information leakage, we resolve this by refining the test set with finer actor boundaries.

## 2 RELATED WORKS

**General Bias Mitigation** Many works have exposed various types biases in machine learning models Suresh & Guttag (2019), finding that not only do they pick up on biases within the training data, but

they amplify them as well Ntoutsi et al. (2020). This can be exceedingly harmful when the bias is related to demographic information, breaking fairness constraints Hardt et al. (2016). Additionally, training on a truly balanced dataset is virtually impossible Wang et al. (2019), and biases existing in a pretrained model tend to transfer into the downstream task Salman et al. (2022). Therefore, it is desirable to seek a solution like ours that mitigates biases at the utility task level instead of at the dataset level. Adversarial training Goodfellow et al. (2014); Xie et al. (2017); Zhang et al. (2018) is a popular method that has proven effective in debiasing neural network representations Beutel et al. (2017); Elazar & Goldberg (2018); Zhang et al. (2018); Wang et al. (2019). These techniques often necessitate the use of supplementary information, such as labels for scenes or objects present. Alternatively, these methods may involve the training of a separate critic model to predict the biased attribute, such as a gender classifier for gendered debiasing Beutel et al. (2017); Wang et al. (2019) or scene/object classifiers Duan et al. (2022); Zhai et al. (2023). They then rely on the negative gradients of the predictor network, making the effectiveness contingent upon the accuracy and reliability of the critic model. In contrast, our adversarial method eliminates the need for specialized knowledge of bias attributes thorough labels or a separate predictor network. We simplify the debiasing process and reduce the computational overhead by utilizing the *same* model with a *different* input.

**Background/Foreground Bias in Action Recognition** Most works exploring bias in action recognition are focused on mitigating the effect of the background representation biases, as models tend to use it over motion information to predict the action Yun et al. (2020); Choi et al. (2019); Wang et al. (2021); Weinzaepfel & Rogez (2021); Ding et al. (2022a); Duan et al. (2022); Byvshev et al. (2022). One approach to mitigate this involves emphasizing learning temporal information over spatial cues. This is often achieved through computationally expensive techniques such as optical flow Sun et al. (2018); Sevilla-Lara et al. (2019) or by decoupling spatial and temporal representations Ding et al. (2022b); Zhang & Crandall (2022). However, these methods suffer from significant computational overhead, complex modeling techniques, or reliance on careful foreground/background mask annotations, limiting their practicality. One promising direction for learning quality temporal features is spatio-temporal contrastive learning Tao et al. (2020); Schiappa et al. (2023); Bahng et al. (2020); Bao et al. (2021), but Wang et al. (2021) notice that in standard formulations, static information still tends to out-compete motion features. In contrast, we enforce a hard constraint on the usefulness of static information, resulting in more strict utilization of temporal motion features. Li et al. (2023) reveals that many existing techniques which are resistant to background biases are still vulnerable to foreground biases. When the foreground video subject is a person, the foreground bias may be related to demographic information, which can cause inappropriate decision making. Taking this into account, our proposed adversarial learning method accounts for *all* spatial information in both the video foreground and background.

## 3 METHOD

The core of ALBAR, our proposed debiasing method, is to properly utilize an adversarial objective that discourages a model from learning to predict action classes based on spatial information alone. The additional proposed losses support this objective and help balance training for optimal performance. Each component of our method is shown in Figure 1. First, we notate the problem and baseline in Section 3.1. Then, we describe the core adversarial loss in Section 3.2. The additional supplementary *entropy maximization* and *gradient penalty* components are described in Section 3.3 and Section 3.4, respectively, with everything put together in Section 3.5.

### 3.1 PROBLEM FORMULATION

A standard bias evaluation setup consists of two decoupled test sets: one that is independent and identically distributed (IID) from the training data, $\mathbb{D}_{IID}$, and another one that is out-of-distribution (OOD) $\mathbb{D}_{OOD}$. Formally, for video action recognition, we have video dataset $\mathbb{D} = \{(\mathbf{x}^{(i)}, \mathbf{y}^{(i)})\}_{i=1}^{N}$, where $\mathbf{x}^{(i)}$ is the $i$th video instance, $\mathbf{y}^{(i)}$ is the associated one-hot action label, and $N$ is the number of samples in the training dataset. A model is trained on $\mathbb{D}$, then evaluated on both unseen test sets $\mathbb{D}_{IID}$ and $\mathbb{D}_{OOD}$. Debiasing methods attempt to learn robust and generalizable features that can maximize performance on $\mathbb{D}_{OOD}$ without sacrificing performance on $\mathbb{D}_{IID}$.

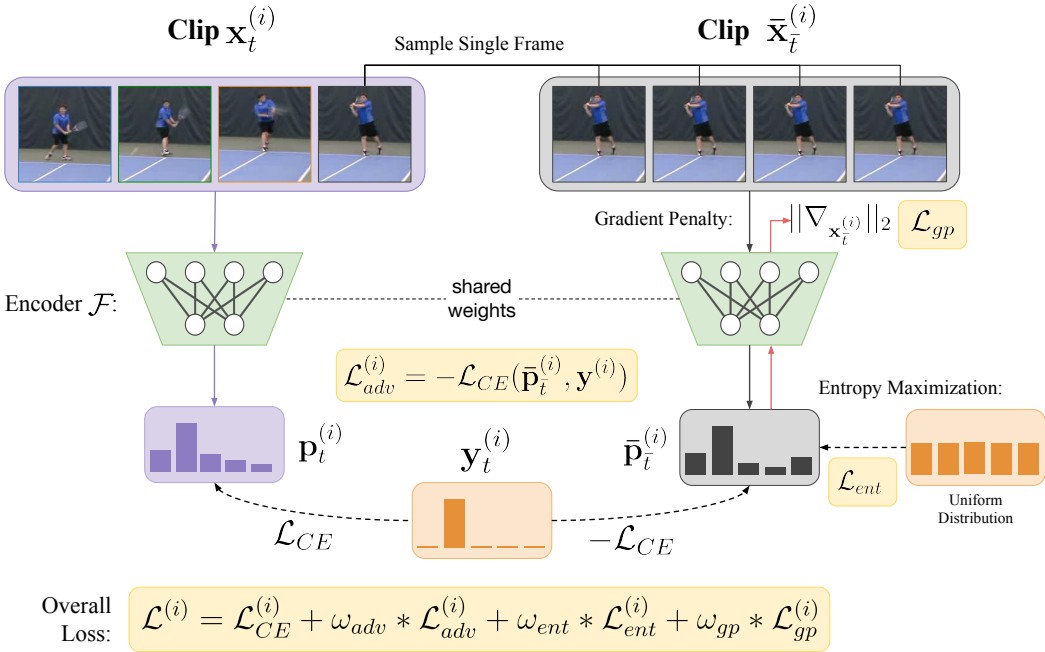

Figure 1: Given video clip $\mathbf{x}_t^{(i)}$, we sample a random frame and stack it to create static clip $\bar{\mathbf{x}}_{\bar{t}}^{(i)}$. Both clips are passed through encoder $\mathcal{F}$ to generate prediction vectors $\mathbf{p}_t^{(i)}$ and $\bar{\mathbf{p}}_{\bar{t}}^{(i)}$. The adversarial loss (Eq. 2) is computed by taking the cross-entropy of the motion clip prediction $\mathbf{p}_t^{(i)}$ and subtracting the cross-entropy of the static clip prediction $\bar{\mathbf{p}}_{\bar{t}}^{(i)}$. This static prediction is encouraged to be uncertain by the entropy loss (Eq. 3), and the gradients related to the prediction (shown in red, Eq. 4) are encouraged to be lower for more stable training by the gradient penalty loss.

**Baseline** The conventional form of supervised video action recognition employs a standard empirical risk minimization (ERM) cross-entropy loss (Eq. 1) to guide model predictions toward the ground truth label distribution $\mathbf{y}$. The formulation is as follows:

$$\mathcal{L}_{CE}^{(i)} = -\sum_{c=1}^{N_C} \mathbf{y}_c^{(i)} \log \mathbf{p}_c^{(i)}, \tag{1}$$

where $N_C$ is the total number of action classes in $\mathbb{D}_{IID}$ and $\mathbf{p}_c^{(i)}$ is the prediction vector. This objective is typically effective at maximizing performance on $\mathbb{D}_{IID}$, but fails to properly generalize to a disjoint $\mathbb{D}_{OOD}$ Duan et al. (2022). Li et al. (2023) demonstrate that this loss allows for static information to erroneously correlate with action labels, leading to static-related biases being used in the final model predictions.

## 3.2 STATIC ADVERSARIAL LOSS

To solve this static bias problem, the correlation between static information and action label needs to be broken. Therefore, we propose an adversarial method to break this correlation by directly discouraging the model's ability to predict the action given a single frame. Over the course of training, the model must learn to utilize primarily temporal information to achieve high performance, as it is discouraged from using spatial information. We hypothesize that reducing reliance on spatial information leads to more robust usage of temporal action information, allowing for reduced bias and better generalization to $\mathbb{D}_{OOD}$. Specifically, given input clip $\mathbf{x}_t^{(i)}$, we sample a single frame at a specified time $\bar{t}$ within $\mathbf{x}_t^{(i)}$, stacking it $F$ times to create static clip, $\bar{\mathbf{x}}_{\bar{t}}^{(i)}$. See Section 4.5 for a discussion on $\bar{t}$ frame selection. This new boring "clip" has zero motion. Both clips are passed

through backbone model $\mathcal{F}$ to generate prediction vectors $\mathbf{p}_t^{(i)} = \mathcal{F}(\mathbf{x}_t^{(i)})$ and $\bar{\mathbf{p}}_{\bar{t}}^{(i)} = \mathcal{F}(\bar{\mathbf{x}}_{\bar{t}}^{(i)})$. The model is trained in such a manner that the prediction $\mathbf{p}_t^{(i)}$ is still aligned to the ground-truth one-hot vector $\mathbf{y}^{(i)}$, but the cross-entropy between the prediction vector of the static clip $\bar{\mathbf{p}}_{\bar{t}}^{(i)}$ and $\mathbf{y}^{(i)}$ is *maximized*. This is accomplished by reversing the gradient of the cross-entropy loss of static-clip $\bar{\mathbf{x}}_{\bar{t}}^{(i)}$. Because these clips share high mutual information, differing only in motion patterns, the model should learn to utilize only the motion patterns to satisfy the loss conditions.

Formally, our adversarial loss objective can be expressed as follows:

$$\mathcal{L}_{adv}^{(i)} = -\mathcal{L}_{CE}(\bar{\mathbf{p}}_{\bar{t}}^{(i)}, \mathbf{y}^{(i)}), \tag{2}$$

where $\bar{\mathbf{p}}_{\bar{t}}^{(i)}$ is the prediction vector of the static clip. This objective creates a strong negative learning signal, discouraging the model from using static information in its classifications. Note that this method does not require the use of any additional labels to achieve debiased representations.

### 3.3 Entropy Maximization

Simply applying the two above objectives Eq. 1 & Eq. 2 alone results in degraded performance (Tab. 3, row b). We find that the model still learns the spatial correlations, but to satisfy the loss conditions, given static clip $\bar{\mathbf{x}}_{\bar{t}}^{(i)}$, it will predict an incorrect class with high confidence. In order to combat this, we propose to directly add an *entropy maximization* loss to the static-clip prediction vector $\bar{\mathbf{p}}_{\bar{t}}^{(i)}$. By encouraging the model to predict all classes with an equal probability given a clip with no motion, it prevents the label flipping trivial solution, forcing the model to rely on the clips with motion to make high-confidence predictions. The entropy loss is formulated as follows:

$$\mathcal{L}_{\text{ent}}^{(i)} = \sum_{c=1}^{N_C} \bar{\mathbf{p}}_{\bar{t},c}^{(i)} \log(\bar{\mathbf{p}}_{\bar{t},c}^{(i)}), \tag{3}$$

where $\bar{\mathbf{p}}_{\bar{t},c}^{(i)}$ is the softmax prediction probability for class $c$. The summation computes the entropy of the predictions over all classes. The general formulation for entropy is negative, but since we want to *maximize* it, we negate the sum again, which gives us the loss function that we can minimize during training. The lower bound of this loss occurs when the prediction distribution is uniform, meaning that all classes have equal probabilities. This way, the encoder is trained to have higher uncertainty when not provided temporal motion information, ideally losing the ability to predict actions properly based on spatial information.

### 3.4 Gradient Penalty

Even though the entropy loss helps prevent a trivial solution training collapse, we find that training is still unstable, experiencing major fluctuations in performance on intermittent validation steps (see Appendix Fig. 6). The objective is for the encoder to learn better temporal representations, not directly react to static inputs and drastically update weights. As such, we need to design a loss that penalizes significant changes in the encoder's output when presented with static inputs. Taking inspiration from stabilizing GAN training Gulrajani et al. (2017), we add a *gradient penalty* loss. Instead of interpolating between samples and pushing the mean gradient norm towards 1, we simplify the formulation by directly minimizing the gradient norm with respect to the static clip $\bar{\mathbf{x}}_{\bar{t}}^{(i)}$. This effectively acts as a regularizer to stabilize the encoder by reducing sensitivity to static input, thereby causing the learning of more robust and generalizable motion features. The proposed gradient penalty loss is defined as follows:

$$\mathcal{L}_{gp}^{(i)} = ||\nabla_{\bar{\mathbf{x}}_{\bar{t}}^{(i)}} \mathcal{F}(\bar{\mathbf{x}}_{\bar{t}}^{(i)})||_2, \tag{4}$$

where $||\cdot||_2$ represents the $\ell_2$ norm and $\nabla_{\bar{\mathbf{x}}_{\bar{t}}^{(i)}} \mathcal{F}(\bar{\mathbf{x}}_{\bar{t}}^{(i)})$ represents the gradients of model $\mathcal{F}$ w.r.t. static clip input $\bar{\mathbf{x}}_{\bar{t}}^{(i)}$. In practice, this objective promotes smoothness and consistency in the adversarial gradients, leading to more stable model convergence (Appendix Fig. 6).

### 3.5 COMBINED TRAINING OBJECTIVE

Putting everything together, our overall training objective is described by taking the average of all samples $(i) \in \mathbb{D}_{IID}$ computed using the following equation:

$$\mathcal{L}^{(i)} = \mathcal{L}_{CE}^{(i)} + \omega_{adv} * \mathcal{L}_{adv}^{(i)} + \omega_{ent} * \mathcal{L}_{ent}^{(i)} + \omega_{gp} * \mathcal{L}_{gp}^{(i)}, \tag{5}$$

where $\omega_{adv}$, $\omega_{ent}$, and $\omega_{gp}$ are the relative weightage of each of the proposed loss functions.

## 4 EXPERIMENTS

### 4.1 DATASETS

**Kinetics400 Carreira & Zisserman (2017)** is a large-scale action dataset that is commonly used to pretrain models for a better initialization on various downstream tasks. However, the dataset has been shown to exhibit bias towards the background information. This can lead models to rely on background cues rather than focusing on the actual human actions.

**UCF101 Soomro et al. (2012)** is a popular mid-sized action recognition dataset. Similar to Kinetics, it displays a high degree of background bias, *i.e.*, the actions can be identified using the background.

**HMDB51 Kuehne et al. (2011)** is a smaller action recognition dataset. Unlike Kinetics400 and UCF101, HMDB51 has a large intra-class background variance, resulting in less reliance on static background information. Conversely, HMDB51 exhibits a relatively high degree of *foreground* bias, *i.e.*, the actions can be identified using static foreground information.

**ARAS Duan et al. (2022)** is a real-world test set based on Kinetics400. It is designed to be OOD for scene-debiasing evaluation by consisting of only examples of rare background scenes for each action.

**SCUBA and SCUFO Li et al. (2023)** are background and foreground bias evaluation benchmarks for action recognition based on common benchmarks Kinetics400, UCF101, and HMDB51. SCUBA replaces the background of action clips with alternate images from the following sources: the test set of Places365 Zhou et al. (2017), generated by VQGAN-CLIP Crowson et al. (2022), or randomly generated stripes following sinusoidal functions. SCUFO takes a frame from a SCUBA video and stacks it into a motionless clip to evaluate potential foreground bias.

### 4.2 IMPROVED UCF101 SCUBA/SCUFO PROTOCOL

Li et al. (2023) proposed SCUBA and SCUFO datasets and metrics to evaluate both background and foreground bias in video action recognition models. These protocols require masks to extract the foreground from the original clips. However, the masks used for the UCF101 variation are bounding boxes from the THUMOS-14 challenge Jiang et al. (2014). Thus, surrounding background information is carried into the bias evaluation videos, as seen in Figure 2. This is not sufficient to evaluate performance on this dataset, as a classifier reliant on the background can still take advantage of this information to score high on the protocol. To mitigate this effect, we utilize a flexible video object segmentation model, SAMTrack Cheng et al. (2023); Yang et al. (2021); Yang & Yang (2022); Kirillov et al. (2023); Liu et al. (2023) to segment the actors (subjects) in each video. The actors are initially grounded using the same THUMOS-14 bounding boxes. Each testing video is manually checked for accurate segmentation. The same dataset creation protocol proposed by Li et al. (2023) is used to create SCUBA and SCUFO variations with these new masks. The improved benchmark no longer includes background information, such as in Figure 2 (a), tightly bounding the human subject as seen in Figure 2 (b). Results in Table 2 show results on our newly created benchmark. See Appendix Sec. C for results on the existing benchmark.

### 4.3 IMPLEMENTATION DETAILS

For all experiments, we use a clip resolution of $224 \times 224$. We follow Li et al. (2023) and use Kinetics400 Carreira & Zisserman (2017) pretrained Swin-T Liu et al. (2022) with 32 frame clips at a skip rate of 2. We adopt the same common augmentations used in Li et al. (2023): random resized cropping and random horizontal flipping. Our chosen optimizer is AdamW Kingma & Ba (2014);

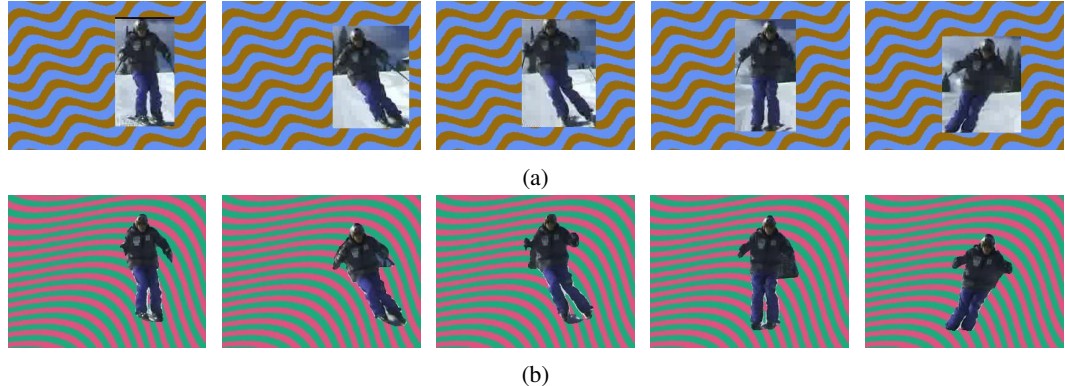

Figure 2: Example clip from UCF101-SCUBA-Sinusoid protocol clip, corresponding to a video from the class "Skiing". (a) shows the frames from previous protocol, where snow is visible in the background. Our improved protocol (b) uses tight segmentation masks to eliminate the background.

Table 1: Comparison results on IID and OOD test sets of various debiasing methods on HMDB51. All experiments use Swin-T pretrained using Kinetics-400. The light blue column highlights the contrasted accuracy.

| Augmentation or Debiasing | IID | OOD | | | |
|---|---|---|---|---|---|
| | | Avg SCUBA ↑ | Avg SCUFO ↓ | Confl-FG ↑ | Contra. Acc. ↑ |
| None | 73.92 | 43.93 | 20.46 | 36.58 | 27.84 |
| Mixup$_{ICLR'18}$ | 74.58 | 43.10 | 21.17 | 36.62 | 26.09 |
| VideoMix$_{arXiv'20}$ | 73.31 | 39.39 | 20.44 | 32.68 | 23.13 |
| SDN$_{NeurIPS'19}$ | 74.66 | 40.02 | 20.22 | 34.87 | 22.88 |
| BE$_{CVPR'21}$ | 74.31 | 43.56 | 19.96 | 35.99 | 27.84 |
| ActorCutMix$_{CVIU'23}$ | 74.05 | 46.79 | 22.07 | 36.97 | 28.12 |
| FAME$_{CVPR'22}$ | 73.79 | 51.40 | 26.92 | 39.61 | 29.66 |
| StillMix$_{ICCV'23}$ | **74.82** | 51.81 | 13.39 | 47.38 | 40.28 |
| Ours | 72.81 | **53.53**↑21.9% | **1.50** ↓92.7% | **48.13** ↑31.4% | **53.22** ↑91.2% |
| Ours w/ StillMix aug. | 74.31 | 54.24 | 1.35 | 47.07 | 53.68 |

Loshchilov & Hutter (2017) with default parameters $\beta_1 = 0.9$, $\beta_2 = 0.999$, and weight decay of 0.01. We follow the linear scaling rule Goyal et al. (2017) with a base learning rate of 1e-4 corresponding to a batch size of 64. For training, we utilize a linear warmup of 5 epochs and a cosine learning rate scheduler. Further details may be found in Appendix Section B.

## 4.4 BACKGROUND/FOREGROUND BIAS EVALUATION

On existing background/foreground bias benchmarks, we follow the usual protocol of reporting the average Top-1 accuracy across 3 runs. We first perform IID evaluation using the original test sets, then evaluate OOD performance using the SCUBA and SCUFO datasets. On the SCUBA protocol, a higher accuracy shows a lower static background bias, yet on the SCUFA protocol, a lower accuracy shows a lower static foreground bias. There is also an additional conflicting foreground version (Confl-FG) that adds a random foreground from one SCUBA video to another in the sinusoidal set. We also report contrasted accuracy Li et al. (2023) (Contra. Acc.), where a video prediction is counted correct if and only if the model is *correct* on a SCUBA video and is *incorrect* on the corresponding SCUFO video. This single score is representative of overall bias reduction performance in both the foreground and the background, and is highlighted in light blue in each table due to its importance. Further explanation of this protocol can be found in Appendix Section A and Figure 4. We compare against previous debiasing action recognition techniques and augmentations Mixup Zhang et al. (2017),

Table 2: Results on IID and OOD test sets of various debiasing methods on UCF101 using our proposed updated protocol. The light blue column highlights the contrasted accuracy. All experiments use Swin-T pretrained using Kinetics-400.

| Augmentation or Debiasing | IID | OOD | | | |
|---|---|---|---|---|---|
| | | Avg SCUBA ↑ | Avg SCUFO ↓ | Confl-FG ↑ | Contra. Acc. ↑ |
| None | 95.51 | 18.78 | 1.50 | 49.36 | 17.53 |
| StillMix ICCV'23 | **96.14** | 24.68 | 0.36 | **55.03** | 24.40 |
| Ours | 94.98 | **26.25 ↑39.8%** | **0.14 ↓90.7%** | 54.02 ↑9.4% | **26.23 ↑49.6%** |
| Ours w/ StillMix aug. | 94.71 | 31.17 | 0.16 | 58.84 | 31.14 |

VideoMix Yun et al. (2020), SDN Choi et al. (2019), BE Wang et al. (2021), ActorCutMix Ding et al. (2022a), and StillMix Li et al. (2023).

The results on the HMDB51 variant are shown in Table 1. Results for the updated UCF101 version can be found in Table 2, with results on the old protocol in Appendix Section C. Remarkably, ALBAR improves upon the previous contrasted accuracy by over **12%** for a total of 53.22% on the HMDB51 protocol. We see that our static adversarial method strongly reduces harmful biases by properly learning robust motion features for video action recognition, without requiring bias attribute specific knowledge. Additionally, as ALBAR employs a combination of training loss objectives and no major augmentations, we can easily combine the StillMix Li et al. (2023) augmentation with our method during training. This pushes the performance further beyond what either method can achieve alone, achieving a contrasted accuracy of 53.68%.

**ARAS Rare Scene Evaluation** Performance on ARAS is evaluated by computing top-1 accuracy using a Kinetics400 trained classifier (the same used in SCUBA/SCUFO protocol). Table 6 in Appendix Section C includes results on both benchmarks. ALBAR is able to properly mitigate real-world scene-related biases, not getting fooled by even extreme cases of rare backgrounds.

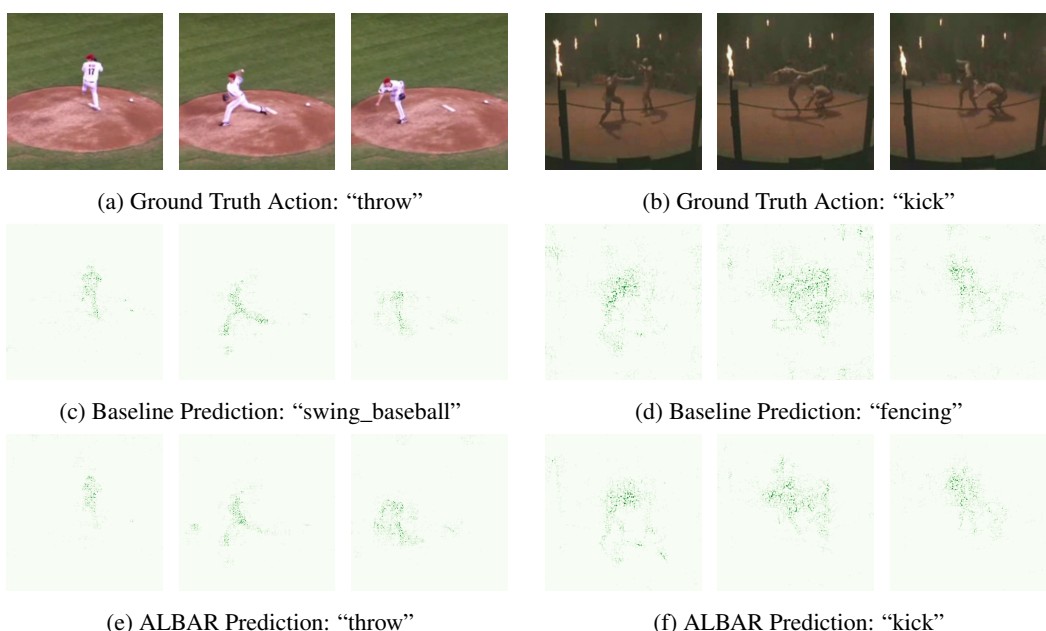

(a) Ground Truth Action: "throw"  (b) Ground Truth Action: "kick"

(c) Baseline Prediction: "swing_baseball"  (d) Baseline Prediction: "fencing"

(e) ALBAR Prediction: "throw"  (f) ALBAR Prediction: "kick"

Figure 3: Qualitative examples from the HMDB51 test set showing the baseline model choosing an incorrect action label due to spatial context. Our method correctly chooses the action label in each. These pixel-level attributions are plotted using integrated gradients Sundararajan et al. (2017)

**Qualitative Results** Figure 3 shows two examples of misclassifications of the baseline model that were correctly interpreted by our debiased model. We plot the pixel-level attributions for the predicted class using integrated gradients Sundararajan et al. (2017) in Figure 3. Interestingly, looking at the baseball_throw example, we see that both the baseline (Fig 3c) and our method (Fig 3e) have similar attribution maps, appropriately looking at the video foreground. The baseline model still chooses the incorrect action class, indicating potential foreground bias, likely using the baseball uniform. In Figure 3b, the camera angle appears similar to instances of fencing, but the correct action is actually "kick". In each of these examples, the biased baseline likely utilizes spurious features in the classification, but ALBAR easily overcomes them.

## 4.5 ABLATIONS AND ANALYSIS

**Training loss components:** Table 3 ablates each individual loss component during debiasing training. Each piece is crucial for achieving maximum performance. The adversarial loss strongly contributes to the foreground debiasing performance by imposing a strong penalty for identifying the correct action utilizing static information. Alone (row b), it causes the model to degenerate into a trivial solution where the model learns the inappropriate static correlations, then intentionally chooses the incorrect class. The entropy maximization objective is useful by itself (row c) in reducing reliance on static information, but is not as strong of an objective as $\mathcal{L}_{adv}$. It combines with the adversarial loss (row e) to prevent the trivial solution and further improve debiasing performance, but training remains unstable (see Appendix Figure 6). The gradient penalty does not have a large effect on its own (row d), but has a positive effect when combined with the other losses, likely because the penalty prevents the static clip gradients from offsetting proper learning from the motion clip gradients. Once all three objectives are utilized simultaneously (row h), we see a strong gain in performance. Each component complements each other well, leading our method to achieve SOTA contrasted accuracy on the SCUBA/SCUFO benchmarks. Additional ablations can be found in Appendix Section C.

Table 3: Ablation to determine the effectiveness of each component loss objective.

| | $\mathcal{L}_{CE}$ | $\mathcal{L}_{adv}$ | $\mathcal{L}_{ent}$ | $\mathcal{L}_{gp}$ | IID | OOD | | | |
| | | | | | | Avg SCUBA ↑ | Avg SCUFO ↓ | Confl-FG ↑ | Contra. Acc. ↑ |
|---|---|---|---|---|---|---|---|---|---|
| (a) | ✓ | ✗ | ✗ | ✗ | 73.92 | 43.93 | 20.46 | 36.58 | 27.84 |
| (b) | ✓ | ✓ | ✗ | ✗ | 72.61 | 42.29 | **0.00** | 32.77 | 42.29 |
| (c) | ✓ | ✗ | ✓ | ✗ | 71.57 | 46.02 | 2.97 | 38.40 | 44.30 |
| (d) | ✓ | ✗ | ✗ | ✓ | 72.09 | 41.13 | 14.69 | 35.16 | 29.24 |
| (e) | ✓ | ✓ | ✓ | ✗ | 72.22 | 46.81 | 0.49 | 41.13 | 46.72 |
| (f) | ✓ | ✓ | ✗ | ✓ | 71.70 | 45.92 | **0.00** | 38.52 | 45.92 |
| (g) | ✓ | ✗ | ✓ | ✓ | **73.86** | 49.14 | 9.74 | 42.89 | 41.11 |
| (h) | ✓ | ✓ | ✓ | ✓ | 72.81 | **53.53** | 1.50 | **48.13** | **53.22** |

**Static frame selection:** Table 4 ablates the frame sampling strategy used to chose $\bar{t}$ for the static clip. Specifically, we evaluate choosing the first, middle, last, and random frame positions. Choosing the first and last frames sharply decreased performance, likely due to scene changes or irrelevant information before and after actions. Contrasting the action learning from irrelevant information is trivial and would not learning anything useful. Random frame selection ensures variety in static objectives, leading to strong results. Notably, middle frame selection improved performance over random selection. This makes sense, since the middle frame is likely to contain the full background and actor in the middle of performing an action, making it a hard negative sample for the adversarial learning process. These findings highlight the importance of choosing a static frame where the action is currently being performed, as it should include both the foreground and background information. While using a sophisticated method to detect actors/backgrounds in frames and choosing static frames based on the presence of both would likely achieve more optimal performance, this adds too much inductive bias and computation, so we leave this up to future work.

Table 4: Ablation to evaluate static frame sampling strategy.

| Frame Sampling Strategy | IID | OOD | | | |
|---|---|---|---|---|---|
| | | Avg SCUBA ↑ | Avg SCUFO ↓ | Confl-FG ↑ | Contra. Acc. ↑ |
| Random | 73.20 | 53.22 | 0.42 | 49.84 | 53.02 |
| First | 72.75 | 50.92 | 0.18 | 45.59 | 50.91 |
| **Middle** | 72.81 | 53.53 | 1.50 | 48.13 | **53.22** |
| Last | 72.68 | 49.49 | 0.40 | 42.42 | 49.40 |

**Downstream task evaluation:** Our results thus far demonstrate the improved capabilities of our method in the constrained problem of trimmed action recognition. Here, we show that debiasing a video encoder leads to performance gains on various downstream video understanding tasks. Specifically, we evaluate on weakly supervised anomaly detection and untrimmed temporal action localization, both of which take in long videos and localize the timestamps where anomalies/specific actions occur. In each of these tasks, feature quality and feature discriminability are crucial to performance. Having an unbiased encoder allows for localized representations to depend on motion rather than the background or biased foreground. For the weakly supervised anomaly detection task, we report results on the UCF_Crime Sultani et al. (2018) dataset as frame-level Area Under the Curve (AUC) of the Receiver Operating Characteristic (ROC), which is the standard evaluation metric for this task. We use one of the current SOTA methods MGFN Chen et al. (2023) with unchanged hyperparameters for this evaluation, only swapping the feature sets used to those extracted from a baseline model and a model trained using our framework on HMDB51. For the temporal action detection task, we report results on the THUMOS14 Jiang et al. (2014) dataset. Evaluation is given as mean average precision (mAP). We use a SOTA model TriDet Shi et al. (2023) with standard hyperparameters, again only swapping the feature sets used in a similar fashion to the UCF_Crime protocol. More details may be found in Appendix Section C. Table 5 highlights results, demonstrating the benefit of our debiasing method across downstream video understanding tasks.

Table 5: Evaluation of our debiased model across downstream anomaly detection (UCF_Crime) and temporal action localization (THUMOS14) tasks.

| Method | Action Recognition HMDB51 (OOD) | Anomaly Detection UCF_Crime | Temporal Action Localization THUMOS14 |
|---|---|---|---|
| Baseline | 27.84 | 82.39 | 54.89 |
| Ours | **53.22** | **84.91** | **55.20** |

**Limitations** Our evaluation focuses on established background and foreground bias protocols, which may not capture all possible real-world scenarios and biases. Expanding the evaluation to include more diverse and realistic settings, including those with harmful demographic biases, would provide a more comprehensive assessment of the practical effectiveness of our method. Additionally, our approach makes an assumption that no static information should be useful. Future work could explore architectures that still eliminate spatial biases while still utilizing the information appropriately.

## 5 CONCLUSION

We propose ALBAR, a novel adversarial training framework for efficient background and foreground debiasing of video action recognition models. The framework eliminates the need for direct knowledge of biased attributes such as an additional critic model, instead leveraging the negative cross-entropy loss of a clip without motion passed through the same model as the adversarial component. To ensure optimal training, we incorporate static clip entropy maximization and gradient penalty objectives. We thoroughly validate the performance of our approach across a comprehensive suite of bias evaluation protocols, demonstrating its effectiveness and generalization across multiple datasets. Moreover, ALBAR can be seamlessly combined with existing debiasing augmentations to achieve performance that significantly surpasses the current state-of-the-art. It is our hope that our work contributes to the development of fair, unbiased, and trustworthy video understanding models.

## 6 ACKNOWLEDGMENTS

This work was supported in part by the National Science Foundation (NSF) and Center for Smart Streetscapes (CS3) under NSF Cooperative Agreement No. EEC-2133516.

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

## APPENDIX OVERVIEW

## A DATASET DETAILS

**Kinetics400 Carreira & Zisserman (2017)** contains approximately 300,000 video clips sourced from YouTube, covering 400 human action classes. It has a single dedicated train/val/test split. In this work, we train on the train split and evaluate IID on the test split.

**UCF101 Soomro et al. (2012)** comprises of 13,320 video clips across 101 action classes and has three train/test splits available. Following Li et al. (2023), we utilize only the first (split 1) train/test split for all training and evaluation in this work.

**HMDB51 Kuehne et al. (2011)** consists of 6,849 video clips covering 51 human activity classes and has three potential train/test splits, much like UCF101. Again, we only use the first (split 1) train/test split for all training and evaluation in this work.

**ARAS Duan et al. (2022)** is only a test set, so all of the 1,038 rare-scene action videos are utilized to compute Top-1 accuracy.

**SCUBA and SCUFO Li et al. (2023)** are also test sets. Each dataset variation contains three variations of different background types to ensure solid generalization to OOD data. All of the data is used to compute Top-1 accuracy, and performance on the related SCUBA and SCUFO splits is combined as described in Main Paper Section 4.4 to compute the contrasted accuracy. Figure 4 includes additional bias evaluation dataset examples.

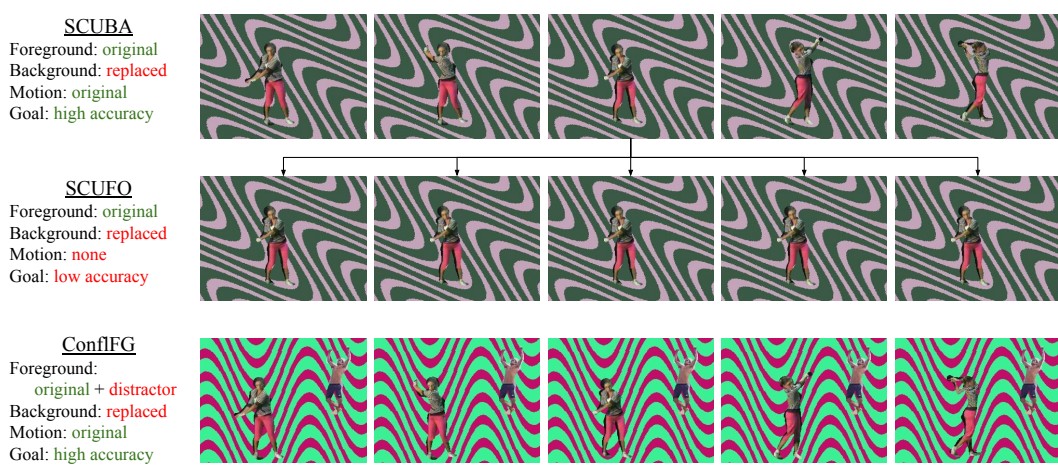

Figure 4: Example SCUBA/SCUFO/ConflFG frames from HMDB51. SCUBA evaluates background bias while SCUFO samples a single frame, evaluating foreground bias. ConflFG adds a static distractor foreground to a SCUBA video, evaluating both bias types simultaneously. Contrasted accuracy requires predicting the correct label on a SCUBA video AND predicting the *incorrect* action on its paired SCUFO video. More information can be found in StillMix Li et al. (2023).

Our proposed fix to the UCF101 protocol is applied across all three background types, namely Places365 images, VQGAN-Clip images, and random sinusoidal images. Further visual examples of the improved protocol are included in Figure 5. Note that in the "Fencing" example, we additionally segment the opposing fencer for a more complete video.

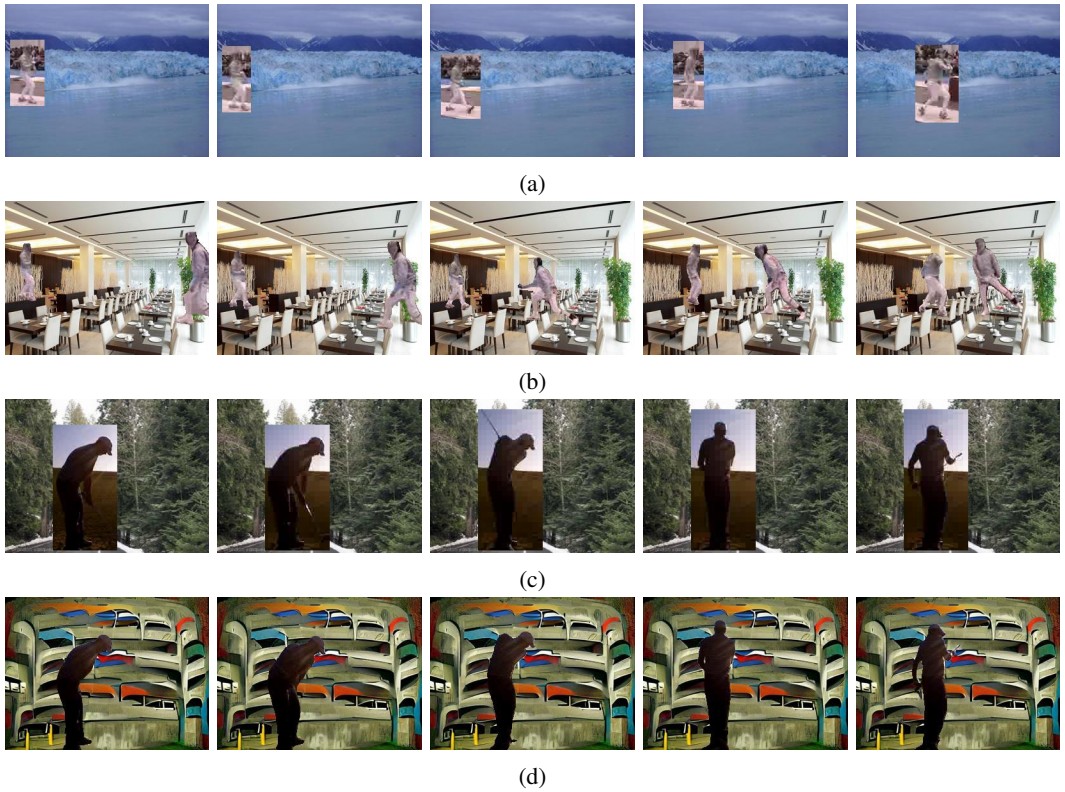

Figure 5: Example clips from UCF101-SCUBA-Places365 and UCF101-SCUBA-VQGAN protocols. (a) shows an example video from the class "Fencing" from the previous protocol. Our improved protocol (b) uses tight segmentation masks to eliminate background information. Likewise, (c) shows an example clip from the class "Golf Swing", and (d) shows the improved segmented version.

## B    IMPLEMENTATION DETAILS

This section is an addition to the details listed in Main Paper Section 4.3. A standard validation set does not exist for HDMB51 and UCF101. We randomly sample 20% of the respective training sets to use for validation, labelling them with an identical process as in 4.2. The PyTorch Paszke et al. (2019) library is utilized for all experiments. All experiments are performed on a local computing cluster with access to V100 and A100 GPUs of various memory configurations up to 80GB. On an 80GB GPU, a batch size of 8 clips is used, and we train for up to 50 epochs. On HMDB51, this may take ≈8 hours with validation every epoch. We follow Li et al. (2023) and evaluate Top-1 accuracy using a single clip sampled from the center of each video. The only augmentations at test time are a resize to short side 256 and a center crop to 224 × 224.

## C    ADDITIONAL EXPERIMENT DETAILS

**Expanded tables:** Table 7 shows expanded (non-averaged) SCUBA/SCUFO results for HMDB51, and Table 8 shows them for the original UCF101 protocol. In the original UCF101 protocol, the test set contains a significant amount of extra background information, leading to skewed results, especially on the SCUBA background bias protocol. The background is mostly replaced, but a good portion of the actual video background remains, inflating scores. Our method does not utilize this well, seemingly performing worse as a result.

As referenced in Main Paper Section 4.4, results on the Kinetics400 and ARAS versions can be found here, in Table 6. Similar outcomes to the other experiments are seen here.

Table 6: Results on IID and OOD test sets of various debiasing methods on Kinetics400. All experiments use Swin-T pretrained using Kinetics-400.

| Augmentation or Debiasing | IID | OOD | | | |
| --- | --- | --- | --- | --- | --- |
| | | Avg SCUBA ↑ | Avg SCUFO ↓ | ARAS ↑ | Contra. Acc. ↑ |
| None | **68.13** | 42.97 | 20.26 | 57.57 | 25.78 |
| StillMix ICCV'23 | 67.27 | **45.83** | 10.72 | 59.21 | 36.60 |
| Ours | 67.58 | 44.75 ↑4.1% | **0.11** ↓99.5% | 59.79 ↑3.9% | **44.74** ↑73.6% |
| Ours w/ StillMix aug. | 68.10 | 46.52 | 0.77 | 60.17 | 46.07 |

Table 7: Quantitative results of various debiasing methods on the HMDB51 Kuehne et al. (2011) dataset. The light blue column highlights the contrasted accuracy. **Bold** and underline indicate best and second best results, respectively.

| Augmentation or Debiasing | HMDB51 | HMDB51-SCUBA (↑) | | | HMDB51-SCUFO (↓) | | | Contra. Acc. (↑) |
| --- | --- | --- | --- | --- | --- | --- | --- | --- |
| | | Places365 | VQGAN-CLIP | Sinusoid | Places365 | VQGAN-CLIP | Sinusoid | |
| None | 73.92 | 47.61 | 42.77 | 41.41 | 20.68 | 17.90 | 22.80 | 27.84 |
| Mixup | 74.58 | 46.70 | 42.49 | 40.12 | 21.25 | 18.47 | 23.78 | 26.09 |
| VideoMix | 73.31 | 41.33 | 38.18 | 38.67 | 19.64 | 18.82 | 22.85 | 23.13 |
| SDN | 74.66 | 41.96 | 40.82 | 37.29 | 19.99 | 19.62 | 21.06 | 22.88 |
| BE | 74.31 | 47.36 | 42.94 | 40.39 | 20.91 | 17.55 | 21.41 | 21.41 |
| ActorCutMix | 74.05 | 50.13 | 46.51 | 43.73 | 22.16 | 20.26 | 23.80 | 28.12 |
| FAME | 73.79 | 54.71 | 53.67 | 45.81 | 27.10 | 27.26 | 26.40 | 29.66 |
| StillMix | **74.82** | 53.27 | 52.43 | 49.73 | 13.39 | 12.66 | 14.13 | 40.28 |
| Ours | 73.20 | **54.73** | **54.80** | **50.12** | **0.04** | **1.17** | **0.04** | **53.02** |
| Ours w/ StillMix aug. | 74.31 | 55.59 | 56.95 | 50.20 | 1.45 | 1.52 | 1.09 | 56.68 |

Table 8: Quantitative results of various debiasing methods on the UCF101 Soomro et al. (2012) dataset, using the existing protocol from Li et al. (2023), which contains background information. The light blue column highlights the contrasted accuracy. **Bold** and underline indicate best and second best results, respectively.

| Augmentation or Debiasing | UCF101 | UCF101-SCUBA (↑) | | | UCF101-SCUFO (↓) | | | Contra. Acc. (↑) |
| --- | --- | --- | --- | --- | --- | --- | --- | --- |
| | | Places365 | VQGAN-CLIP | Sinusoid | Places365 | VQGAN-CLIP | Sinusoid | |
| No | **96.21** | 37.63 | 34.37 | 54.94 | 3.48 | 3.02 | 10.82 | 36.82 |
| Mixup | 96.17 | 39.82 | 40.89 | 57.79 | 2.88 | 3.28 | 11.62 | 40.46 |
| VideoMix | 96.00 | 28.59 | 37.36 | 58.26 | 7.81 | 11.40 | 20.60 | 29.37 |
| SDN | 95.76 | 34.78 | 32.56 | 50.40 | 2.21 | 1.42 | 5.30 | 36.42 |
| BE | 96.06 | 39.76 | 36.16 | 56.01 | 3.55 | 2.93 | 10.15 | 38.62 |
| ActorCutMix | 95.87 | 51.02 | **55.28** | **69.53** | 8.00 | 8.43 | 19.32 | 46.87 |
| FAME | 95.81 | 40.62 | 44.56 | 37.54 | 5.74 | 6.50 | 6.84 | 35.14 |
| StillMix | 96.02 | **55.22** | 53.68 | 65.75 | 2.40 | 2.16 | 5.76 | **54.90** |
| Ours | 94.98 | 46.04 | 42.36 | 55.70 | **0.68** | **0.59** | **0.34** | 47.88 |
| Ours w/ StillMix aug. | 94.79 | 57.19 | 54.78 | 60.91 | 0.13 | 0.31 | 0.22 | 57.58 |

**Downstream task evaluation details:** For the downstream task evaluation presented in Main Paper Table 5, we utilize official implementations from MGFN Chen et al. (2023)[1] for anomaly detection and TriDet Shi et al. (2023)[2] for temporal action localization. The only difference between the baseline and our method is the extracted feature set input to the task models. The baseline uses a Swin-T model finetuned on HMDB51 with the standard cross-entropy objective. Our ALBAR trained

---

[1]https://github.com/carolchenyx/MGFN
[2]https://github.com/dingfengshi/TriDet

encoder corresponds to the model shown in our best reported results on HMDB51 finetuning. The exact provided hyperparameters are utilized and unchanged from the original implementations.

**Method Generalization Example – Image Classification Debiasing:** Our method was built based on an observation of a semi-unique property of video – the fact that a single frame from along the temporal dimension contains all the same 2D information as a 3D clip, but it does not contain the information necessary to classify the end result: an action taking place across time. At the core of this is the idea that we have paired inputs: one containing all necessary information for classification, and one containing mostly the same information, but lacks crucial information to complete the task. Given this unique problem setup in another setting, our methodology should apply. As such, we have put together a quick experiment to evaluate our method in an alternate, albeit related, domain: image classification. Similar to action recognition, the background bias problem is well-known and well studied. In the Waterbirds Sagawa et al. (2019) dataset (based on CUB-200-2011 Wah et al. (2011)), images are built by taking segmented images of specific bird types and placing them on specific background types (land birds, waterbirds vs. land-based, water-based backgrounds), causing models to learn the spurious correlation between the bird type and background type. We observe a similar phenomenon to our problem setup, where instead of having the temporal dimension to reduce, we can reduce information in the spatial dimensions by removing the foreground. Here, our pairing becomes the original image (containing the foreground bird for classification) and the original background image (with no bird, so it should be useless for bird classification). Acquiring these pairs in a non-synthetic setup takes more effort than our video-based setup, but our core method should nonetheless still apply. The results in Table 9 indicate that there is merit to our method outside of video understanding. Our ALBAR method improves both minority classes and worst-group accuracy without spending time to optimize hyperparameters. While this is not a robust analysis, we believe that in setting up these two scenarios (video debiasing, image classification debiasing), we demonstrate the broader applicability of our method across domains, contingent upon having the unique paired input setup.

Note: Since the standard Waterbirds dataset does not separately contain references to the background images used, we had to create a split (using the public code provided by the original authors), modifying it to additionally save the background images to create our pairing.

Table 9: Generalization experiment using Waterbirds Sagawa et al. (2019). Per-class accuracy (%) evaluation is provided, with worst-group accuracy commonly used as an evaluation metric. WoW = Waterbirds on Water, LoL = Landbirds on Land, etc.

| Method | WoW (majority) | WoL (minority) | LoL (majority) | LoW (minority) |
|---|---|---|---|---|
| Baseline (R18) | 92.68 | 50.00 | **99.16** | 76.45 |
| Ours (R18) | **92.99** | **59.50** | 98.67 | **78.00** |

**Effect of gradient penalty:** Figure 6 highlights the efficacy of our additional gradient penalty objective in stabilizing training and improving performance. It prevents the model from taking large steps in different directions as the static adversarial and temporal cross-entropy objectives fight back and forth, leading to overall smoother learning and improved overall performance.

**Strength of adversarial component** $\omega_{adv}$**:** The adversarial loss weight ($\omega_{adv}$) has the most significant impact on training efficacy. Our empirical findings suggest that the optimal value for $\omega_{adv}$ is 1, which equates to the cross-entropy weight of the motion clip. Suboptimal values lead to two contrasting issues: (1) $\omega_{adv} < 1$ insufficiently mitigates static biases, while (2) $\omega_{adv} > 1$ dominates the learning process, overshadowing the crucial information contained within the motion clips. Striking the perfect balance is key to achieving effective training and ensuring the model learns from both the adversarial loss and the motion clips in a complementary manner.

**Strength of entropy maximization** $\omega_{ent}$**:** The entropy objective coefficient plays a critical role in model stability. Inadequate values lead to two distinct failure modes: (1) an insufficiently low weight allows the classifier to misclassify static inputs by assigning incorrect predictions, while (2) an excessively high coefficient impairs the classifier's capacity to capture and learn temporal dependencies effectively. Much like the adversarial loss, balancing the entropy maximization objective's strength is essential for the model to achieve stable and accurate performance.

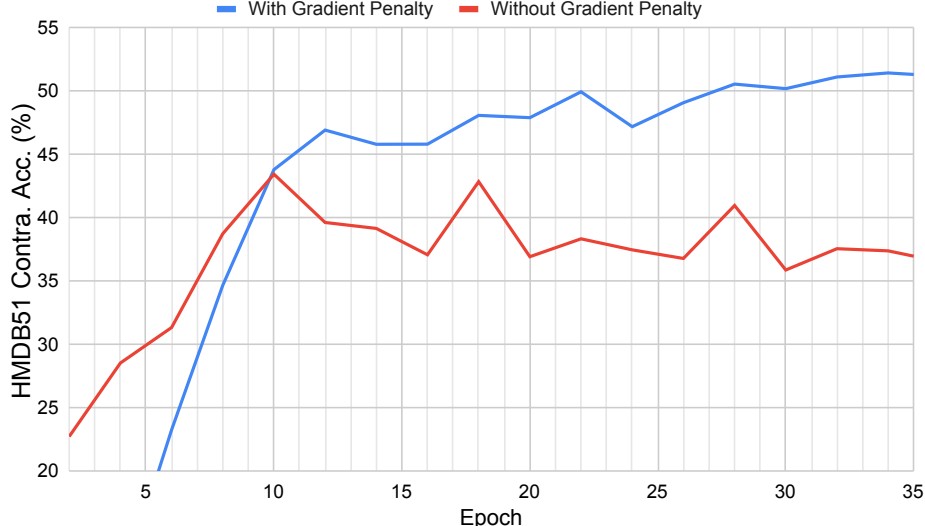

Figure 6: Stability of performance during training with and without the proposed gradient penalty objective.

Table 10: Ablation with adversarial loss $L_{adv}$ weightage.

| $\omega_{adv}$ | IID | OOD | | | |
|---|---|---|---|---|---|
| | | Avg SCUBA↑ | Avg SCUFO↓ | Confl- FG↑ | Contra. Acc.↑ |
| 0 | 73.92 | 43.93 | 20.46 | 36.58 | 27.84 |
| 0.5 | 73.79 | 48.65 | 3.31 | 43.28 | 46.56 |
| 1 | 73.20 | **53.20** | **0.42** | **49.84** | **53.02** |
| 2 | 67.84 | 44.52 | 0.99 | 40.86 | 44.45 |
| 4 | 51.50 | 27.76 | 5.86 | 24.02 | 27.76 |

**Strength of gradient penalty $\omega_{gp}$:** Similar to the other objectives, stable training relies on finding an optimal loss weight $\omega_{gp}$. Ineffective weights lead to two failure cases: (1) a low $\omega_{gp}$ renders the penalty insufficient for acting as a proper regularizer, while (2) a higher coefficient can overshadow the primary training objective and cause the gradients to become oversmoothed, resulting in reduced performance on $\mathbb{D}_{IID}$.

Table 11: Ablation with entropy loss $L_{ent}$ weight.

| $\omega_{ent}$ | IID | OOD | | | |
|---|---|---|---|---|---|
| | | Avg SCUBA↑ | Avg SCUFO↓ | Confl- FG↑ | Contra. Acc.↑ |
| 0 | 73.92 | 43.93 | 20.46 | 36.58 | 27.84 |
| 2 | 72.61 | 51.71 | 0.00 | 44.57 | 51.71 |
| 4 | 73.20 | **53.20** | **0.42** | **49.84** | **53.02** |
| 8 | 74.12 | 49.87 | 1.86 | 43.01 | 49.26 |

Table 12: Ablation with gradient loss $L_{gp}$ weight.

| $\omega_{gp}$ | IID | OOD | | | |
|---|---|---|---|---|---|
| | | Avg SCUBA↑ | Avg SCUFO↓ | Confl- FG↑ | Contra. Acc.↑ |
| 0 | 73.92 | 43.93 | 20.46 | 36.58 | 27.84 |
| 5 | 73.59 | 49.77 | 0.03 | 44.53 | 49.77 |
| 10 | 73.20 | **53.20** | **0.42** | **49.84** | **53.02** |
| 20 | 71.76 | 50.55 | 3.03 | 43.55 | 48.84 |

