# OpenReview forum: "ALBAR: Adversarial Learning approach to mitigate Biases in Action Recognition"
_ICLR.cc/2025/Conference — ICLR 2025 Poster_

### Official Review · Reviewer_aFRh · 2024-10-31

**Soundness:** 1
**Presentation:** 1
**Contribution:** 1
**Rating:** 6
**Confidence:** 5

**Summary:**

This paper presents ALBAR, an adversarial learning method aimed at reducing biases in action recognition models. The study focuses on addressing background and foreground biases, which can impact the performance of applications such as autonomous vehicles and assisted living monitoring. ALBAR utilizes an adversarial training technique to minimize the model's dependence on static background cues, encouraging the use of dynamic motion information for action classification.

**Strengths:**

S1: This paper resents state-of-the-art results on established SCUBA/SCUFO background/foreground debiasing benchmarks, and formulation is technically sound.

S2: Several ablations show the contribution of each component.

**Weaknesses:**

W1: Although the results demonstrate that the model outperforms state-of-the-art methods, the technical contribution is incremental and purely based on previously introduced techniques.

W2: How is the fairness of the comparative experiments ensured in this paper, and how are the results of the comparative methods obtained?

W3: The ALBAR performs well on specific bias evaluation protocols, but its generalization capabilities to new types of biases or different domain tasks have not been fully validated.

W4: Although the paper proposes a simplified end-to-end training framework, adversarial training often involves additional computational costs. The paper does not discuss the computational efficiency and scalability of the ALBAR method in detail.

**Questions:**

Although the results demonstrate that the model outperforms state-of-the-art methods, the technical contribution is incremental and purely based on previously introduced techniques.

How is the fairness of the comparative experiments ensured in this paper, and how are the results of the comparative methods obtained?

The ALBAR performs well on specific bias evaluation protocols, but its generalization capabilities to new types of biases or different domain tasks have not been fully validated.

Although the paper proposes a simplified end-to-end training framework, adversarial training often involves additional computational costs. The paper does not discuss the computational efficiency and scalability of the ALBAR method in detail.

---

> ### Author Response · Authors · 2024-11-22
> **Initial Rebuttal (Part 1/2)**
>
> Thank you for your insightful review and detailed questions.
>
> **_W1: Although the results demonstrate that the model outperforms state-of-the-art methods, the technical contribution is incremental and purely based on previously introduced techniques._**
> - **Response to Weakness & Question 1:** While the basic idea of adversarial training, entropy maximization, and gradient penalties are not alone novel, we want to highlight the aspects of our technical contribution that are novel:
>     1. First, adversarial training typically utilizes labels or separate models to facilitate the counter-objective, while we reduce complexity and create a strong negative by utilizing the same encoder and classifier head, manipulating the input itself instead. By consolidating adversarial training techniques into a single, streamlined framework, we offer a more efficient and integrated approach to addressing bias in video action recognition.
>         - Previous works have utilized separate 2D and 3D encoders, repelling representations in a contrastive-like objective. We are the first to combine the objective into the same encoder and adversarially train in this manner.
>     2. Our formulation of minimizing gradient norm to stabilize adversarial training weight updates w.r.t. a specific input type represents a novel technical contribution. This approach provides a more nuanced method of managing adversarial training dynamics.
>
> **_W2: How is the fairness of the comparative experiments ensured in this paper, and how are the results of the comparative methods obtained?_**
>
> - **Response to Weakness & Question 2:** Great care was taken to ensure fairness in experimentation. Most results were sourced from prior publications. The StillMix paper released their codebase along with comprehensive hyperparameter choices. Due to this, we were able to replicate their numbers in our own implementation.  We retained their original hyperparameter choices and incorporated our proposed losses, ensuring a fair comparative evaluation. In the case of results not being reported previously—on our proposed UCF101 protocol fix—we used the reproduced model for evaluation.
>
> **_W3: The ALBAR performs well on specific bias evaluation protocols, but its generalization capabilities to new types of biases or different domain tasks have not been fully validated._**
>
> - **Response to Weakness & Question 3:** We appreciate your point here. In the following Table 1, we demonstrate our method's generalizability across different domain tasks (weakly supervised anomaly detection, temporal action localization). Regarding biases, the SCUBA/SCUFO protocols are designed to comprehensively address static biases in video action recognition, notably introducing foreground bias evaluation. Currently, we are unaware of alternative protocols for assessing other types of video action recognition bias. We welcome suggestions for additional bias evaluation methods and would be eager to incorporate them.
> In the table, HMDB51 (OOD) refers to the Contrasted Accuracy (Contra. Acc.) results explained in Main Paper Section 4.4. The downstream tasks use features extracted from our debiased model trained on HMDB51. Notably, performance is greatly improved across tasks that require high-quality temporal understanding. These results highlight that our approach is not only relevant for action recognition but also beneficial for diverse downstream tasks in video understanding, further establishing its broader impact. For additional details on this implementation/analysis, please refer to our initial response to Reviewer `EAVn`.
>
> Table 1: Additional video understanding task results comparison between baseline model and one trained with our ALBAR framework. Action recognition performance is reported as Top-1 accuracy (%), anomaly detection score is given as AUC (%), and temporal action detection as mAP (%). A higher score is desired for both tasks.
> | Method   |  HMDB51 (OOD) | UCF_Crime | THUMOS14 |
> | :---:    |  :---:     | :---:     | :---:     |
> | Baseline |  27.84     | 82.39     | 54.89     |
> | Ours     |  **53.22** | **84.91** | **55.20** |

---

> > ### Comment · Reviewer_aFRh · 2024-12-02
> >
> > After reading the rebuttal, I chang my rating from 5 to 6.

---

> ### Author Response · Authors · 2024-11-22
> **Initial Rebuttal (Part 2/2)**
>
> **_W4: Although the paper proposes a simplified end-to-end training framework, adversarial training often involves additional computational costs. The paper does not discuss the computational efficiency and scalability of the ALBAR method in detail._**
>
> - **Response to Weakness & Question 4:** This is a good point for us to address. While adversarial training does introduce computational overhead, our framework is intentionally designed to minimize additional computational costs. Specifically, the only additional cost comes from creating the static video batch and passing it through the model in addition to the standard clips, which scales only linearly with the video encoder size. Critically, we avoid loading additional models, introducing new parameters, or overly complex adversarial architectures. On HMDB51, using a single 80GB A100 GPU, our complete training process (including validation each epoch) requires approximately 9 hours—demonstrating the method's computational efficiency. With the only additional overhead coming from the computation of extra losses coming from the same model, our approach is lightweight and practical. Apart from the training computational cost, it is important to note that our method incurs no additional computational overhead during practical deployment compared to a standard model.

---

### Official Review · Reviewer_EAVn · 2024-11-04

**Soundness:** 3
**Presentation:** 3
**Contribution:** 2
**Rating:** 6
**Confidence:** 4

**Summary:**

The paper proposes a method to improve generalization capability of an activity recognition method in temporally-segmented video datasets.  The method focuses on an adversarial approach to removing biases from static elements of the scene.  The paper takes a random frame from the video and creates a "static" video by repeating the frame the same length as the original clip, then using this static clip as the adversary.  The paper combines this adversarial approach with other reasonable loss terms.  It demonstrates state of the art accuracy on the problem.

**Strengths:**

- The problem of bias mitigation in ML is seriously important.  The paper proposes a concrete approach to mitigating clear bias problems in action recognition.

- The entropy maximization term makes sense for avoiding the static cues.

- The paper uses practical mechanisms to overcome the challenges of training the models.

- The evaluation is performed on a recent OOD benchmark for bias analysis, and performs well relative to other methods.

- The writing is mostly clear and concrete.

**Weaknesses:**

- The paper is is somewhat "small" in that although it describes what seems to be a new method and its evaluation on reasonable benchmarks, there is little discussion around key elements of the method, their limitations, their rationale, etc.  (see questions)
EDIT: The review discussion thus far, and the other reviewers seems to have identified similar concerns.  Discussion around this point suggests there is some generality to visual problems, but some concern about the narrowness of the contribution remains.  Most of the comparative papers are in CV conferences.  I raised my scored, but this point should be discussed among the AC pair/triplet however ICLR is doing it this round.

- Does the "static clip" way of approaching video bias mitigation, have broader utility in video understanding or beyond?

- Certain interesting experiments are omitted, even though the writing suggests they may be useful.  For example, L233 reads "A naive application of Eq. 2 results in degraded performance."  What exactly is a naive application?  Furthermore, this implies that the two parts of Eq. 2 is individually interesting; in particular, does the right hand side do anything?  The ablation study does not include these two parts separately.

**Questions:**

- The mechanism for adversarial learning in the paper is very specific to action recognition.  How can this mechanism be generalized to be of broader interest to the ICLR community?

- The method works by sampling any frame from the video and repeating it for a static clip.  Why any frame?  Aren't certain frames better or worse than others for the stated goal?  (The pre-segmented nature of the datasets in question create itself an algorithmic bias.  In the stated real world application deployments no such pretemporal segmentation is avavilable, and hence brings into question the feasability of the method in practice.)  But, more concretely to the task, why is there no analysis whatsoever on the impact of this frame selection?  Even for a subset of one dataset, it would have been interesting to understand the breadth of potential with different static clips.

- Wouldn't it be clearer to concretely specify that the $\vby$ action label notation is a one-hot vector?  It is one-hot, right?

- At line 218, shouldn't "maximized" be "minimized"?  At least, something in that sentence does not match up: "p(t) is still matched to gt distribution y, but the similar ....maximized"  Maximizing the "similarity" (a loose term here) is minimizing the ce loss.

- How are there reasonable IID results in Table 3 for the case that Ladv is not used ---> when it is not used, there is no actual gradient to guide the model to do any recognition?

---

> ### Author Response · Authors · 2024-11-22
> **Initial Rebuttal (Part 1/4)**
>
> Thank you for highlighting our strengths and for the comprehensive analysis of the paper.
>
> **_W1: The paper is is somewhat "small" in that although it describes what seems to be a new method and its evaluation on reasonable benchmarks, there is little discussion around key elements of the method, their limitations, their rationale, etc. (see questions)_**
> **_Q1: The mechanism for adversarial learning in the paper is very specific to action recognition. How can this mechanism be generalized to be of broader interest to the ICLR community?_**
> - **Response to Weakness 1 & Question 1:** We humbly request the reviewer to consider the significance of action recognition as a foundational problem in the broader domain of video understanding. Below, we provide both context and empirical evidence to demonstrate the broader applicability of our approach:
>
>     - **Importance of Action Recognition in Video Understanding:**
>         - Action recognition serves as the core problem for advancements in video understanding. Many state-of-the-art models in the diverse tasks of video understanding are trained on large-scale action recognition datasets (e.g., Kinetics-400) and subsequently adapted to diverse tasks. Even though standard action recognition models are trained on trimmed videos, their learned representations are directly transferable to various downstream tasks, often without fine-tuning (frozen pretrained model). Such tasks require an encoder to compute high-quality local action understanding, then slide this across videos and model global information across these sets of local low-dimensional features, instead of trying to pass in the high-dimensional videos all at once. In these scenarios, having an unbiased, powerful encoder for trimmed action recognition is crucial.
>     - **Empirical Evidence of Broader Applicability of our Method:**
>         - We provide empirical results in the table below to demonstrate the utility of our proposed debiasing training paradigm across multiple downstream tasks:
>             - **Anomaly Detection:** The first downstream task we evaluate on is weakly supervised anomaly detection. The task requires a model to localize the frames within long, untrimmed videos where some anomaly (defined by the dataset) occurs. Due to the long videos, this task typically starts with a set of features extracted in sliding-window fashion from a Kinetics400 (or similarly) pretrained video encoder. In this task, discriminability between feature segments is crucial in order to localize anomalous segments. If an encoder exhibits a background bias for example, then it may output similar features for two clips with the same background but drastically different foregrounds, making the anomalous segments difficult to distinguish. For this weakly supervised anomaly detection task, we report results on the UCF_Crime [1] dataset as the frame-level Area Under the Curve (AUC) of the Receiver Operating Characteristic (ROC), which is the standard evaluation metric for this task. We use one of the current SOTA methods MGFN [3] with unchanged hyperparameters for this evaluation, only swapping the feature sets used to those extracted from a baseline model and a model trained using our framework on HMDB51.
>             - **Temporal Action Localization:** The second additional task we evaluate on is untrimmed temporal action localization. The goal of this task is to identify time intervals where particular action classes occur within long video. Similar to weakly supervised anomaly detection, this tasks utilizes features extracted from a pretrained video encoder and stands to benefit from models with less static bias and better temporal modeling. For the temporal action detection task, we report results on the THUMOS14 [2] dataset. Evaluation is given as mean average precision (mAP). We use a SOTA model TriDet [4] with standard hyperparameters, again only swapping the feature sets used in a similar fashion to the UCF_Crime protocol.
>
>
>
>         - In the table, HMDB51 (OOD) refers to the Contrasted Accuracy (Contra. Acc.) results explained in Main Paper Section 4.4. Table 1 (next comment, Part 2) shows all of these results. Notably, we see that performance is greatly improved across tasks that require high-quality temporal understanding. These results highlight that our approach is not only relevant for action recognition but also beneficial for diverse downstream tasks in video understanding, further establishing its broader impact.

---

> > ### Author Response · Authors · 2024-11-22
> > **Initial Rebuttal (Part 2/4)**
> >
> > Table 1: Additional video understanding task results comparison between baseline model and one trained with our ALBAR framework. Action recognition performance is reported as Top-1 accuracy (%), anomaly detection score is given as AUC (%), and temporal action detection as mAP (%). A higher score is desired for both tasks.
> > | Method   |  HMDB51 (OOD) | UCF_Crime | THUMOS14 |
> > | :---:    |  :---:     | :---:     | :---:     |
> > | Baseline |  27.84     | 82.39     | 54.89     |
> > | Ours     |  **53.22** | **84.91** | **55.20** |
> >
> >
> > ### Citations
> > [1] Waqas Sultani, Chen Chen, and Mubarak Shah. Real-world anomaly detection in surveillance videos. In Proceedings of the IEEE conference on computer vision and pattern recognition, pp. 6479–6488, 2018.
> > [2] Y.-G. Jiang, J. Liu, A. Roshan Zamir, G. Toderici, I. Laptev, M. Shah, and R. Sukthankar. THUMOS challenge: Action recognition with a large number of classes. http://crcv.ucf.edu/ THUMOS14/, 2014.
> > [3] Yingxian Chen, Zhengzhe Liu, Baoheng Zhang, Wilton Fok, Xiaojuan Qi, and Yik-Chung Wu. Mgfn: Magnitude-contrastive glance-and-focus network for weakly-supervised video anomaly detection. In Proceedings of the AAAI Conference on Artificial Intelligence, volume 37, pp. 387–395, 2023.
> > [4] Dingfeng Shi, Yujie Zhong, Qiong Cao, Lin Ma, Jia Li, and Dacheng Tao. Tridet: Temporal action detection with relative boundary modeling. In Proceedings of the IEEE/CVF Conference on Computer Vision and Pattern Recognition, pp. 18857–18866, 2023.

---

> > ### Comment · Reviewer_EAVn · 2024-11-22
> > **Discussion (what the author's called "Initial Rebuttal" Part 1 and 2 / 4)**
> >
> > The response unfortunately does not address the questions directly.
> > (1) The question does not doubt the importance of action recognition in video understanding.   The many thousands of papers on the topic over the last 15 years is sufficient to do that.  The question states:  "there is little discussion around key elements of the method, their limitations, their rationale, etc."  Where in the paper is that discussion, that analysis?
> >
> > (2) Broader applicability.  The review comment stated "The mechanism for adversarial learning in the paper is very specific to action recognition. How can this mechanism be generalized to be of broader interest to the ICLR community?"  The response noted anomaly detection and temporal action localization.  Indeed these are downstream tasks in video understanding.  However, that is not the essence of the question, which perhaps was a bit unclear, my mistake.  Let me rephrase more directly.  "How is the proposed adversarial learning method useful and relevant to problems outside of video understanding, if it is?"

---

> > > ### Author Response · Authors · 2024-11-25
> > > **Discussion**
> > >
> > > We apologize for the missed response, we had misinterpreted your concerns. We had thought that your concern was with the limited scope of specifically action recognition, but we see now that you are referring to how the core methodology could be applicable outside of video understanding as a whole. Throughout the method and ablation section, we discuss rationale for adding each component due to limitations of each on their own, combining for the best performance. However, to your point, the analysis is indeed limited to our direct problem setting, not necessarily a general analysis of the components.
> > >
> > > Our paper scope is positioned as an improvement on previous background and foreground debiasing works in action recognition, similar to papers like StillMix [1], Background Erasing (BE) [2], & ActorCutMix [3]. That being said, it would certainly strengthen the work to look outside of video understanding and express how the method components may be of use to the general community. Our method was built based on an observation of a semi-unique property of video -- the fact that a single frame from along the temporal dimension contains all the same 2D information as a 3D clip, but it does not contain the information necessary to classify the end result: an action taking place across time. At the core of this is the idea that we have paired inputs: one containing all necessary information for classification, and one containing mostly the same information, but lacks crucial information to complete the task. We believe that given this unique problem setup, our methodology should apply. As such, we have put together a quick experiment to evaluate our method in an alternate, albeit related, domain: image classification. Similar to action recognition, the background bias problem is well-known and well studied. In the Waterbirds [4] dataset (based on CUB-200-2011 [5]), images are built by taking segmented images of specific bird types and placing them on specific background types (land birds, waterbirds vs. land-based, water-based backgrounds), causing models to learn the spurious correlation between the bird type and background type. We observe a similar phenomenon to our problem setup, where instead of having the temporal dimension to reduce, we can reduce information in the spatial dimensions by removing the foreground. Here, our pairing becomes the original image (containing the foreground bird for classification) and the original background image (with no bird, so it should be useless for bird classification). Acquiring these pairs in a non-synthetic setup takes more effort than our video-based setup, but our core method should nonetheless still apply. The results in Table 3 below indicate that there is merit to our method outside of video understanding, seeing as we improve both minority classes and worst-group accuracy, even without spending time to optimize hyperparameters. While this is not a robust analysis, we believe that in setting up these two scenarios (video debiasing, image classification debiasing), we demonstrate the broader applicability of our method across domains, contingent upon having the unique paired input setup.
> > >
> > > Table 3: Generalization experiment using Waterbirds [4]. Per-class accuracy (%) evaluation is provided, with worst-group accuracy commonly used as an evaluation metric. WoW = Waterbirds on Water, LoL = Landbirds on Land, etc.
> > > | Method | WoW (majority) | WoL (minority) | LoL (majority) | LoW (minority) |
> > > | :--- | :---: | :---: | :---: | :---: |
> > > | Baseline (R18) | 92.68 | 50.00 | **99.16** | 76.45 |
> > > | Ours (R18) | **92.99** | **59.50** | 98.67 | **78.00** |
> > >
> > >
> > > We would like to emphasize that the scope of this work was meant to advance the current state of foreground and background debiasing in action recognition, but we thank you for recognizing that our methods are potentially useful to the general ICLR community, not just those working in video understanding. It is our hope that publishing our work, along with our preliminary analysis into applications in new domains, would expose these ideas to the ICLR community, giving them the opportunity for future exploration of similar methods in their respective domains.
> > >
> > > Note: Since the standard Waterbirds dataset does not separately contain references to the background images used, we had to create a split (using the public code provided by the original authors), modifying it to additionally save the background images to create our pairing.

---

> > > > ### Author Response · Authors · 2024-11-25
> > > >
> > > > #### Citations:
> > > > [1]  Haoxin Li, Yuan Liu, Hanwang Zhang, and Boyang Li. Mitigating and evaluating static bias of action representations in the background and the foreground. In Proceedings of the IEEE/CVF International Conference on Computer Vision, pp. 19911–19923, 2023.
> > > >
> > > > [2]  Jinpeng Wang, Yuting Gao, Ke Li, Yiqi Lin, Andy J Ma, Hao Cheng, Pai Peng, Feiyue Huang, Rongrong Ji, and Xing Sun. Removing the background by adding the background: Towards background robust self-supervised video representation learning. In Proceedings of the IEEE/CVF Conference on Computer Vision and Pattern Recognition, pp. 11804–11813, 2021.
> > > >
> > > > [3]  Shuangrui Ding, Maomao Li, Tianyu Yang, Rui Qian, Haohang Xu, Qingyi Chen, Jue Wang, and Hongkai Xiong. Motion-aware contrastive video representation learning via foregroundbackground merging. In Proceedings of the IEEE/CVF Conference on Computer Vision and Pattern Recognition, pp. 9716–9726, 2022a.
> > > >
> > > > [4]  Sagawa, Shiori, et al. "Distributionally robust neural networks for group shifts: On the importance of regularization for worst-case generalization." arXiv preprint arXiv:1911.08731 (2019).
> > > >
> > > > [5]  C. Wah, S. Branson, P. Welinder, P. Perona, and S. Belongie. The Caltech-UCSD Birds-200-2011 dataset. Technical report, California Institute of Technology, 2011.

---

> > > > > ### Comment · Reviewer_EAVn · 2024-11-25
> > > > >
> > > > > Thank you for the more direct response to the statement about the generality of the proposed method.  The discussion is relevant and fine.  I'll update the review to match my understanding of the paper and how it may fit into ICLR.

---

> ### Author Response · Authors · 2024-11-22
> **Initial Rebuttal (Part 3/4)**
>
> **_W2: Does the "static clip" way of approaching video bias mitigation, have broader utility in video understanding or beyond?_**
>
> - **Response to Weakness 2:** Video biases, such as foreground or background bias, primarily stem from the static appearance cues in a video. These biases arise due to the spurious correlation between the appearance and the action label, leading to predictions that rely on appearance rather than motion. By using a static clip, which inherently eliminates motion information, we ensure that the video encoder generates features solely based on the static appearance of the frames. Penalizing the model for making predictions based on these static appearance features helps mitigate appearance-based biases, including foreground and background biases. Since static clips exclusively capture appearance-related information, this approach can be broadly useful in mitigating all types of biases that originate from appearance, extending its utility beyond just action recognition to other video understanding tasks.
>
> **_W3: Certain interesting experiments are omitted, even though the writing suggests they may be useful. For example, L233 reads "A naive application of Eq. 2 results in degraded performance." What exactly is a naive application? Furthermore, this implies that the two parts of Eq. 2 is individually interesting; in particular, does the right hand side do anything? The ablation study does not include these two parts separately._**
> **_Q5: How are there reasonable IID results in Table 3 for the case that Ladv is not used ---> when it is not used, there is no actual gradient to guide the model to do any recognition?_**
> - **Response to Weakness 3 & Question 5:** This confusion appears comes from our non-ideal notational choices for Ladv In all experiments, we include the cross entropy loss on the standard motion clip (the left hand side of Ladv as written). In cases where Ladv is not applied, technically this just means that omega_adv = 0 (the right hand side weight). We grouped these components into a single equation to illustrate their adversarial relationship, acknowledging this approach complicates future notation, though we acknowledge that this makes future notation tricky. To improve clarity, we propose separating Ladv into distinct losses and will adjust notation accordingly throughout our work. This modification will help prevent misinterpretation and make the mathematical formulation more transparent.
>     - **Weakness 3 part 1:** Naive application may not be the best wording here. This simply means applying the loss (Ladv) without the additional regularizing losses (Lent, Lgp).
>     - **Weakness 3 part 2:** Technically, the ablation that evaluates the left hand side separately from the right hand side of Ladv is Main Paper Table 3 row (a) vs. row (b). We did not ablate using just the static adversarial component without the base temporal cross-entropy, though doing so just results in the model devolving into a state where it predicts the same class for every input, technically achieving 1.96% on IID HMDB51. Addressing notation should clarify this point.
>     - **Question 5:** To directly address Q5, the baseline L_CE is still utilized in situations where Ladv is not, just not the right hand static adversarial side (omega_adv = 0).

---

> ### Author Response · Authors · 2024-11-22
> **Initial Rebuttal (Part 4/4)**
>
> **_Q2: The method works by sampling any frame from the video and repeating it for a static clip. Why any frame? Aren't certain frames better or worse than others for the stated goal? (The pre-segmented nature of the datasets in question create itself an algorithmic bias. In the stated real world application deployments no such pretemporal segmentation is avavilable, and hence brings into question the feasability of the method in practice.) But, more concretely to the task, why is there no analysis whatsoever on the impact of this frame selection? Even for a subset of one dataset, it would have been interesting to understand the breadth of potential with different static clips._**
>
> - **Response to Q2:** This is a very interesting point. We wanted to ensure that our method did not require any additional input or assumptions about what biases were occurring to maintain solid generalization and minimize computation. At first, this meant not having a method/model to choose frames that are more biased than others. However, this is needlessly pedantic, since choosing a frame position from within the clip takes minimal input and can have a drastic effect. As you say, it is interesting to evaluate the impact of different static clips. We explored frame selection strategies for static clips, evaluating first, middle, last, and random frame positions, shown in Table 1 below.
> Here is a summary of our key findings:
>     - First and last frames sharply decreased performance, likely due to scene changes or irrelevant information before and after actions, even in the trimmed setting.
>         - Contrasting the action learning from irrelevant information is trivial, not learning anything useful.
>     - Random frame selection ensures variety in static objectives, leading to strong results.
>     - Notably, middle frame selection improved performance over random selection!
>         - This makes sense, since the middle frame is likely to contain the full background and actor in the middle of performing an action, making it a hard negative sample for the adversarial learning process.
>
>     Using a sophisticated method to detect actors/backgrounds in frames (in trimmed or untrimmed setting) and choosing frames based on the existence of both in the chosen static frames would likely achieve the best performance, but this adds too much inductive bias and computation, so we avoid this in this work. Thanks for bringing this up, this experiment led to valuable insights that improved our overall results.
>
>     Table 1: Experiment with choosing a specific frame for the static adversarial objective.
>     | Method        | IID   | SCUBA | SCUFO | ConflFG | ContraAcc |
>     | :---          | :---: | :---: | :---: | :---: | :---: |
>     | Random Frames | 73.20 | 53.22 |  0.42 | 49.84 | 53.02 |
>     | First Frame   | 72.75 | 50.92 |  0.18 | 45.59 | 50.91 |
>     | Middle Frame  | 72.81 | 53.53 |  1.50 | 48.13 | **53.22** |
>     | Last Frame    | 72.68 | 49.49 |  0.40 | 42.42 | 49.40 |
>
>
> **_Q3: Wouldn't it be clearer to concretely specify that the action label notation is a one-hot vector? It is one-hot, right?_**
>
> - **Response to Q3:** You are correct, this the label vector is technically a one-hot vector, we can specify this in the final version.
>
> **_Q4: At line 218, shouldn't "maximized" be "minimized"? At least, something in that sentence does not match up: "p(t) is still matched to gt distribution y, but the similar ....maximized" Maximizing the "similarity" (a loose term here) is minimizing the ce loss._**
>
> - **Response to Q4:** Thanks for reading closely and for pointing this out, this is a crucial part to get correct in writing. As you say, the loss should be maximized, not the "similarity" (perhaps alignment is a better term). It should be more clear to say that the cross-entropy should be maximized in this paragraph.

---

> > ### Comment · Reviewer_EAVn · 2024-11-22
> > **Discussion response to Parts 3 and 4**
> >
> > These are answered together because they are related.
> >
> > The answers here seem reasonable.    The additional experimental data on the frame selection is interesting, and relevant to help understand the paper.  But, I do not see this in the current pdf.  It would have been helpful to understand how the paper actually evolves based on this discussion (and the other one above).

---

### Official Review · Reviewer_5f7A · 2024-11-08

**Soundness:** 3
**Presentation:** 3
**Contribution:** 3
**Rating:** 6
**Confidence:** 3

**Summary:**

This paper proposes a novel adversarial learning-based method to mitigate biases in action recognition, which provides simplified end-to-end training and does not require any labels/classifiers for bias-related attributes.

**Strengths:**

+ This paper is well-written and well-organized.
+ Good performance on the popular action recognition datasets.

**Weaknesses:**

This paper introduces adversarial learning into action recognition, which is a relatively novel idea, but I have some concerns as follows.

- Are there more visual examples that can depict biases in action recognition?
- Whether the method proposed in this paper can be used for skeleton-based action recognition task？

**Questions:**

See weaknesses

---

> ### Author Response · Authors · 2024-11-22
> **Initial Rebuttal**
>
> Thank you for your positive comments!
>
> **_W1: Are there more visual examples that can depict biases in action recognition?_**
>
> - **Response to Weakness 1:** Absolutely, we can include more. Please download the updated supplementary material. We have added a video file titled `paper7609_bias_examples.mp4`, more qualitative examples can be seen there. Let us know if you would like to see any more.
>
> **_W2: Whether the method proposed in this paper can be used for skeleton-based action recognition task？_**
>
> - **Response to Weakness 2:** While we have not explicitly validated our methodology in a skeleton-based context, our core approach would definitely transfer. One potential bias in skeleton action recognition could be related to static poses, for example predicting a "throwing" action based on a single pose with one arm raised. Applying our static adversarial loss could mitigate this static pose orientation bias, encouraging the model to better consider the motion differences between skeletons. This could be an exciting future direction, and we plan to publicly release our code to support and encourage further exploration in this area.

---

### Meta-Review · Area_Chair_9g1v · 2024-12-17

**Metareview:**

This paper proposes a novel adversarial learning-based method to mitigate biases in action recognition, which provides simplified end-to-end training and does not require any labels/classifiers for bias-related attributes. This paper is well-written and well-organized. Good performance on the popular action recognition datasets. Reviewers are concerned about the scalability and generalization of the proposed methods, the incremental technical contribution, and the fairness of method comparison. After rebuttal, the author has addressed these major concerns. The final vote is acceptance.

**Additional Comments On Reviewer Discussion:**

After the rebuttal period, the reviewers raised the vote to accept. So this submission is above the acceptable standard.

---

### Decision · Program_Chairs · 2025-01-22

Accept (Poster)